# LEARNING TO REASON ON HARD PROBLEMS WITH PRIVILEGED ON-POLICY EXPLORATION

## ABSTRACT

Reinforcement learning (RL) has improved the reasoning abilities of large language models (LLMs), yet state-of-the-art methods cannot use all training problems in a training dataset. On-policy RL rarely produces even a single correct rollout on hard problems, yielding no reward signal or learning altogether. Moreover, mixing easy problems into the training set can detrimental as on-policy RL may derive a larger signal to sharpen its distribution from these problems, impairing its ability to solve harder problems reliably. While one might attempt to address this by distilling human- or model-written solutions into models, these traces are not only expensive and hard to write, but also serve as poor fine-tuning targets: while they produce correct outputs, these concise paths are extremely challenging to learn from. We introduce **Privileged On-Policy Exploration** (`POPE`), a framework that leverages already available solutions from humans or other models to obtain a learning signal on hard problems by using them as "privileged" information that guides exploration. Concretely, `POPE` augments hard prompts with a minimal solution prefix as guidance, enabling RL to obtain non-zero rewards when rolling out conditioned on this prefix. We show that this approach allows RL to acquire behaviors that transfer back to original problems. This process expands the set of solvable problems and improves performance on challenging reasoning benchmarks.

## 1 INTRODUCTION

Reinforcement learning (RL) has substantially improved the reasoning abilities of large language models (LLMs) in math and coding. For example, relatively small models (under 2B parameters) trained with RL to make best use of test-time compute by running longer chains of thought (CoT) can outperform much larger models trained without RL (Setlur et al., 2025b; Liu et al., 2025). Concurrently, some studies also argue that RL post-training mainly amplifies capabilities already present in the model (Yue et al., 2025; Zhao et al., 2025), though others show design choices (prompt mixtures, token budgets, curricula) mitigate such concerns (Setlur et al., 2025b; Liu et al., 2025).

However, even the most effective RL methods fail to train on the full set of available problems, leaving substantial performance gains untapped. This is largely because on-policy RL cannot sample even a single rollout with a non-zero outcome reward on a sizable fraction of "hard" problems, and thus receives no reward signal[1]. In many such cases, rollouts never employ the right strategy, so no correct samples are obtained at all. Procedures such as dynamic prompt sampling (Yu et al., 2025; Wang et al., 2025b) even explicitly filter these prompts out. As a result, RL plateaus once the easier problems are solved. In fact, our results show that after sufficiently many easy problems yield reward, RL simply "sharpens" its behavior on these problems, which reduces the diversity of the model's outputs and impairs its ability to solve new problems compared to the base model. We explain this phenomenon through the lens of ray interference (Schaul et al., 2019), a phenomenon where on-policy RL exhibits a bias of maximizing reward more on states where it already attains reward.

Our goal in this paper is to design an approach that allows RL to overcome this interference issue and learn on hard problems. If the base model is unable to sample any correct rollout on these problems, how can we obtain learning signal though? A natural idea is to collect "expert" reasoning traces from an oracle (e.g., human), either for distillation (Sessa et al., 2024; Agarwal et al., 2024a) or

---

[1]Our analysis shows that when fine-tuning Qwen3-4B on DAPO-MATH-17K (Yu et al., 2025), the model samples a correct rollout on $<50\%$ of prompts, given $K = 32$ parallel attempts at each and 16k token budget.

directly in RL (Yan et al., 2025). Yet reasoning traces of the kind required for LLMs are prohibitively expensive to write, and prior work finds little benefit from using available human-written data (and we corroborate these results). Even when distillation on reasoning traces from bigger models succeeds, gains are marginal (Yan et al., 2025) and upper bounded by the capabilities of bigger models. Therefore, we ask: is there a less-constraining way to use available sources of privileged information such as human-written solutions to derive learning signal on hard training problems?

Our key insight is that privileged information can effectively guide an LLM's *on-policy* exploration on hard problems, even when it is not useful as a training target. For instance, consider a hard problem where the LLM repeatedly pursues incorrect approaches and fails within the allocated training budget. Supplying even a short prefix of a human-written solution can significantly increase the likelihood of reaching the correct answer. This effect is especially useful when the base model already has strong instruction-following capabilities, allowing it to understand and build upon the privileged content. Our approach, **Privileged On-Policy Exploration** (`POPE`), leverages this principle to guide exploration in RL on hard problems, serving as an alternative to distillation, SFT, or off-policy RL.

Concretely, for any set of hard problems, `POPE` first gathers a human- or oracle-provided solution. It then trains a base LLM with RL on a prompt mixture that includes both the original hard problems and versions augmented with partial prefix of these solutions. These partial solutions provide just enough guidance to make it possible to sample at least *one* correct rollout among many attempts. Training on this mixture enables RL to sample some non-zero reward on hard problems, though only when the partial solution is present. Through on-policy exploration guided by these augmented prompts, the model acquires useful behaviors that transfer back to the original prompts, demonstrating generalization (Figure 1; blue). We also find that it

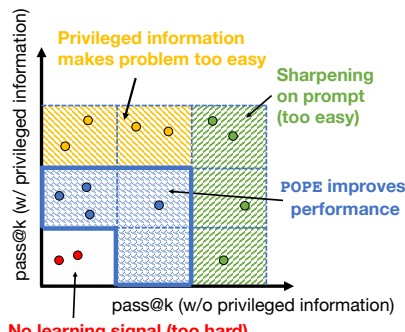

Figure 1: *Illustrating prompts as point in the 2D-plane based on their accuracy.* On prompts with low pass@$k$ but high pass@$k$ under privileged conditioning, `POPE` helps by training on a mixture of normal and augmented prompts. On easy prompts, standard RL already sharpens the model and conditioning may not help, while on prompts that become too easy under conditioning (high pass@$k$ with conditioning but low without), `POPE` is less effective since transfer of behaviors does not occur. Thus, `POPE` uses the smallest prefix of privileged information to guide on-policy exploration to allow maximal transfer on hard prompts, in effect staying within the blue region in the figure.

is critical to ensure that the privileged information provided is such that it does not make the problem substantially simpler, or else this inhibits transfer (Figure 1; yellow). In contrast, standard on-policy exploration or RL applied after distilling human-written solutions fails to solve new problems during training. Empirically, we find that `POPE` enables models to solve hard training problems that remain unsolvable with standard RL training by using either human solutions or language-model generated solutions. Beyond improving pass@1 accuracy, `POPE` also boosts pass@k, showing that it prevents RL from collapsing into merely sharpening the distribution on problems the model already solves.

## 2 RELATED WORK

Our work tackles exploration on hard problems in RL-trained reasoning models when naïvely scaling on-policy RL compute makes little progress. It is related to prior works on enhancing exploration with different training objectives or those that use off-policy traces to reinforce the model being trained and prevent the training from stalling. We briefly review these below.

**RL exploration limits on hard problems.** Small RL-trained models trained via RL can now outperform much larger base models (Liu et al., 2025; Luo et al., 2025), largely by reinforcing long chain-of-thought traces that exhibit meta-strategies and behaviors such as self-correction (Qu et al., 2024) and reflection (Gandhi et al., 2025). Yet, without careful design, RL often *sharpens* the base model's distribution, reducing exploration diversity and leaving hard problems underexplored. This often manifests as a degrading pass@$k$ compared to the base model (Yue et al., 2025; Zhao et al., 2025). To mitigate this drop, one line of work focuses on regularizing RL training to prevent over-sharpening. Examples include adding exploration bonuses based on intrinsic motivation (Gao et al., 2025), entropy (Wang et al., 2025b), count-based bonuses (Song et al., 2025), or training objectives that directly optimize pass@$n$ (Chow et al., 2024; Balashankar et al., 2025). While effective at stabilizing learning, these methods remain constrained by sparse rewards, reliance on easy

problems for signal, and persistent failure on challenging tasks (He et al., 2024). A complementary line of work (Setlur et al., 2025b) amplifies exploratory asymmetries of the base model, such as the verification-generation gap (Setlur et al., 2025a; Song et al., 2024), to generate longer traces beyond the base distribution. Negative gradients in RL can chain such asymmetries over iterations (Zhu et al., 2025), but models often "under-think" (Wang et al., 2025c), executing incorrect high-level strategies that persist despite more rollouts. Our work tackles this by conditioning on privileged information to guide new strategies, discover correct rollouts, and internalize them without explicit conditioning, thereby overcoming the exploration bottleneck.

**Updating LLMs with off-policy traces.** When on-policy exploration struggles due to sharpening or over-thinking, several works propose supervising the RL policy on human- or oracle-provided traces (Lightman et al., 2023; Corrado et al., 2024). However, for methods that rely on traces from a teacher model, the gains are inherently bounded by teacher capacity (Agarwal et al., 2024b). Moreover, learning from such traces typically requires additional techniques such as reward shaping (Yan et al., 2025), entropy control (Wang et al., 2025a), and extensive hyperparameter tuning Zhang et al. (2025). A more fundamental limitation is that off-policy reasoning traces are simply not available for many hard problems: although human-written solutions exist for nearly any RL training prompt and can be rephrased into more effective formats, producing long chains of thought that align with how models actually reason is much more difficult. As we show in our experiments, learning from off-policy data under such type mismatch often leads to a collapse in the diversity of responses sampled by the model. An approach that avoids using off-policy data as training targets is therefore preferable, and our method falls into this category. Related directions include conditioning on subgoals or plans (Hong et al., 2025), or providing abstractions (Qu et al., 2025), but our approach is substantially simpler since it directly conditions on a prefix of an oracle solution. The work most closely related to ours is training with on-policy exploration on adaptively revealed solutions (Amani et al., 2025). However, unlike us, this work focuses on non-reasoning didactic domains where short responses suffice and exploration is easier. A key aspect of our motivation leverages the parameterization of reasoning traces and their strong instruction-following capabilities, which are absent from this work.

## 3 Preliminaries and Notation

We situate this paper in the domain of RL post-training of large languagge models (LLMs) on math problems. For any given problem $\mathbf{x} \sim \rho$ and a rollout $\mathbf{y} \sim \pi_\theta(\cdot \mid \mathbf{x})$ attempting to solve this problem, we define a *binary outcome reward* $r(\mathbf{x}, \mathbf{y}) \in \{0, 1\}$ indicating correctness of the final answer produced by the rollouts. Analogous to work studying RL with 0/1 rewards, we assume that the rollout $\mathbf{y}$ represents the final answer in a \boxed{} block. We study several measures of performance, including the pass@$k$ metric, given byL pass@$k = \Pr\left[\exists y_1, ..., y_k \sim \pi_\theta(\cdot \mid x) \text{ s.t. } r(x, y_i) = 1\right]$, which measures the probability that at least one of $k$ independent attempts at the problem succeeds. This metric captures the role of parallel exploration during training and governs whether a batch can yield any positive signal for GRPO (Shao et al., 2024) or any other Monte-Carlo rollout based policy optimization algorithm for LLMs. We also use pass@$k$ as an evaluation metric to understand the optimization behavior of RL on the base LLM, especially in regard to interference and sharpening effects as we will see in the next section. Most RL algorithms use the *policy gradient*:

$$\nabla_\theta J(\theta) = \mathbb{E}_{y \sim \pi_\theta}\left[r(\mathbf{x}, \mathbf{y}) \nabla_\theta \log \pi_\theta(\mathbf{y} \mid \mathbf{x})\right]. \tag{1}$$

which reinforces rollouts that end in a correct final answer. This process is also called outcome-reward RL. In practice, some of the most-commonly used RL algorithms such as GRPO, uses a reference policy $\pi_{\text{old}}$ for sampling and normalize rewards into *advantages* before utilizing them in the policy gradient: $A_i(\mathbf{x}, \mathbf{y}_i) = r(\mathbf{x}, \mathbf{y}_i) - \frac{1}{n}\sum_{j=1}^{n} r(\mathbf{x}, \mathbf{y}_j)$, so that updates depend on deviations of reward from the batch mean. This normalized structure makes RL brittle on hard problems. If all $n$ rollouts fail on a given problem $\mathbf{x}$ ($r(\mathbf{x}, \mathbf{y}_i) = 0$), then the advantage for all samples vanishes, $A_i = 0$, and the gradient update is exactly zero on $\mathbf{x}$. Thus when pass@$k \approx 0$, training stalls: GRPO cannot generate signal, even with large batch sizes or extended iterations. As we discuss in Section 4, this creates a feedback loop where the model sharpens on easy problems but halts learning on hard ones.

## 4 Ray Interference in RL Post-Training

To motivate our approach, we begin by studying the dynamics of RL training. When training with RL, we often observe that average rewards improve steadily on the training dataset. These rewards are computed by averaging over multiple rollouts for each problem (Shao et al., 2024), which means

a higher average reward can arise either from producing more correct rollouts on an easy problem or from producing at least one correct rollout per problem. A natural hypothesis is that the policy first learns to solve the easier problems in the mixture within a few RL updates, generating multiple correct rollouts per problem. One might then expect that longer training would eventually lead to success on the harder problems as well. However, we find that this does not occur. As shown in Figure 2, a typical RL training run first reduces the fraction of problems unsolved by the base model, but once some problems are solved, the percentage of fully solved problems increases (see step 100 in Figure 2, right), while the model's ability to solve previously unsolved problems drops.

In fact, after step 150, even some problems that were solvable by the base model before become unsolvable by the RL-trained model (the "% unsolvable problems" increases in Figure 2). Training continues to make progress on fully solving a different subset of problems, but this pattern, representative of a typical RL fine-tuning run, highlights that once RL trains models to solve some easy prompts, further training actively inhibits progress on other prompts. The most natural explanation is that this stems from a form of update interference across prompts, known as ray interference in the RL literature (Schaul et al., 2019), as we discuss below.

Figure 2: *Solvable/unsolvable problems in a typical RL run.* **Left**: Percentage of problems a model fails to solve within 16 rollouts (% Unsolvable) first decreases, suggesting the model initially learns to solve new problems, but later increases, indicating that the model has lost its ability to solve solve problems it previously could. **Right**: Percentage of problems the model solves perfectly (% Perfectly Solvable) steadily increases, suggesting that gains in reward come mainly from distribution sharpening rather than solving truly new problems. Moreover, as the model learns to perfectly solve some problems, it loses its ability to solve others.

*Ray interference: why learning from hard problems get harder during RL.*

Typical RL algorithms for post-training LLMs explore on-policy: for each prompt in the batch, we sample $n$ parallel rollouts from the current policy and use them to compute a policy gradient update. If at least one rollout solves the problem but not all of them succeed, the update reinforces the subset of correct rollouts. If none of the $n$ rollouts succeed, or if all of them succeed, then no update is applied on that prompt. Exploration is therefore wasteful when the model cannot sample any correct trace on a hard problem across the $n$ parallel attempts. However, as soon as the policy begins to reliably solve easier problems, their successful rollouts dominate the reward signal. Because all prompts share the same model weights, gradient updates that improve easy problems can inadvertently reduce the probability of exploring useful directions on harder ones since the gradient on easy problems encourages the model to hone in on the right answers and "sharpen" its probability distribution. Over time, this creates a "rich-gets-richer" dynamic: prompts with some correct rollouts get reinforced further (Figure 2, Right), while prompts with no successes become increasingly unsolvable. This corroborates pass@$n$ declining in RL from prior work, despite more rollouts (Yue et al., 2025).

> **Takeaways: Ray interference progressively hurts exploration on hard problems**
>
> - As RL starts solving some problems, its ability to solve other problems degrades.
> - This manifests as a U-shaped trend in the percentage of unsolved problems during training, and a rich-gets-richer effect where prompts get disproportionately improved upon in RL.

## 5 POPE: PRIVILEGED ON-POLICY EXPLORATION

In this section, we develop our approach for training LLMs on hard problems, thereby avoiding the issue outlined above. To address the limitations of on-policy exploration, we make use of privileged data (e.g., human solutions or LLM solutions) to enable experiencing some positive reward on such problems. A natural way to exploit this data is through distillation of oracle traces. However, this strategy can be problematic: LLMs solve hard problems via reasoning traces, which introduces a "type mismatch" with the human or oracle data available. Indeed, we find in our experiments, fine-tuning on oracle data naïvely reduces output diversity on hard problems (Figure 8) without meaningfully improving success rates on the same or related tasks. We therefore build an alternative approach called POPE, which utilizes two main motivating insights in this section.

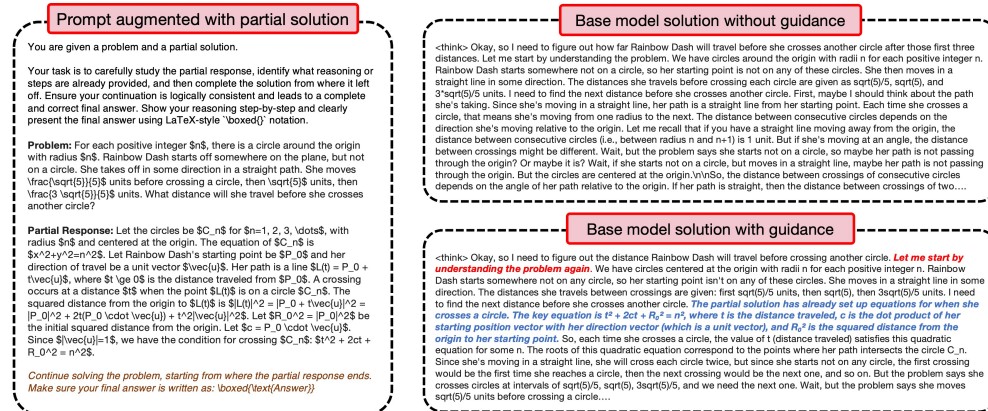

**Figure 3:** *An example of rollout generated by Qwen3-4B base model on a hard training problem* when conditioning and not conditioning on partial oracle solution (guidance). Observe that the rollout with guidance not only refers back to the solution, but it recalls the partial solution in its reasoning, restating many of the facts, and continues the solution from there. This confirms the motivation behind the design of POPE.

## 5.1 MOTIVATING INTUITIONS BEHIND POPE

*1) Conditioning guides better exploration when the base model exhibits strong instruction-following capabilities.* Our first main motivating intuition informing the design of POPE is that an effective way to use privileged information that also avoids distillation is to condition the base model on the prefix of an oracle solution along with the prompt for training. The system prompt then asks the model to complete the solution after carefully analyzing the provided partial oracle solution. This process reduces the difficulty of the problem for the base model in many ways. For instance, some partial solutions might already tackle a *subproblem*, leaving a fewer number of subproblems to be solved for the model's rollout (Figure 3); some other partial solutions may verbalize a plan to tackle the problem, which keeps the model on track (Figure 10). In summary, prefixes of oracle solutions can provide useful privileged information for reducing problem complexity, and address the issue with

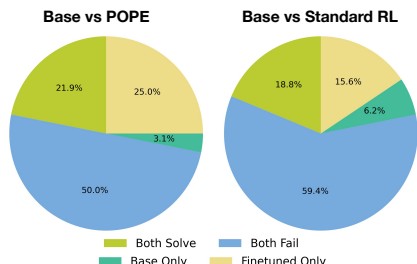

**Figure 4:** *Fraction of problems solved by POPE and standard RL* (evaluated without conditioning on privileged information) compared to the base model, normalized over all problems where conditioning the base model on privileged information yields at least one successful rollout. POPE solves more problems than standard RL, showing that conditioning on privileged information enables it to transfer successful rollouts obtained with conditioning on these training problems to the setting when no conditioning is provided.

cloning target solutions. Given a base model with strong instruction- following abilities, conditioning in this way allows the model to sample correct traces by following the information provided by the partial oracle solution (Figure 3) or by tackling a simpler version of the problem that remains unsolved after the partial solution is provided. As a result, the model can obtain non-zero reward on augmented versions of otherwise unsolvable hard problems. Our approach, POPE prescribes training on versions of problems augmented with partial attempts, as we discuss below.

*2) Training on augmented and normal prompts transfers to successful rollouts on unaugmented prompts.* While the above mechanism allows us to obtain reward on augmented versions of hard prompts, it is not obvious whether training on these prompts with partial solutions leads to effective *transfer* to the original unaugmented prompts. Our second motivation is that when models use a reasoning format that encloses computation within "think" blocks and reiterates the problem statement (and the privileged information in the case of augmented prompts; see Figure 3), training on augmented prompts should often suffice to induce non-zero success on the original problems without privileged guidance. We attribute this to the conditional next-token distributions reinforced early on during training on augmented prompts: reasoning traces on augmented prompts rephrase the problem and reproduce the privileged information, and get reinforced if they experience reward. When the model is presented with unaugmented prompts, it first rephrases the problem statement, and the conditional next-token distributions learned on augmented prompts then help fill in the missing privileged content and guide the remainder of the solution. In effect, learning to solve the augmented prompt also teaches the model how to approach the unaugmented version through a form of a "gluing

mechanism", inherent to reasoning parameterizations. We present an example rollout illustrating this mechanism later in Figure 10. Quantitatively, Figure 4 shows the fraction of problems solved by POPE, standard RL, and the base model without conditioning, out of all problems solvable by the base model when given privileged information. Training on both augmented and normal prompts enables POPE to solve more of these problems compared to RL without requiring conditioning, which shows that training on a mixture of prompts transfers to unaugmented prompts.

## 5.2 THE POPE ALGORITHM

We now present a concrete algorithm based on the insights developed in the previous section. Instead of using human or oracle-written solutions as training targets, we condition on partial prefixes of these solutions as privileged information to generate on-policy rollouts. Formally, given an oracle solution $\mathbf{z}$ to a hard training problem $\mathbf{x} \sim \mathcal{D}_{\text{hard}}$, we condition rollouts on a prefix $\mathbf{z}^{0:i}$ of $\mathbf{z}$, i.e., $\mathbf{y} \sim \pi(\cdot|\mathbf{x}, \mathbf{z}^{0:i})$. In principle, any prefix $\mathbf{z}^{0:i}$ could be used, but we only need the minimal prefix that allows on-policy rollouts to obtain *some* non-zero reward on $\mathbf{x}$, thereby driving learning on $\mathbf{x}$. We therefore choose the shortest prefix that yields non-zero reward under the base model for every prompt $\mathbf{x}$. Formally, for a given prompt $\mathbf{x}$ and base model $\pi_\theta^{\text{base}}$, the minimal privileged prefix is:

$$i^*(\mathbf{x}) := \arg\min_i \ \left\{ i \in [0, L_\mathbf{x}] \ : \ \mathbb{E}_{\mathbf{y} \sim \pi_\theta^{\text{base}}(\cdot|\mathbf{x}, \mathbf{z}^{0:i})} \left[ r(\mathbf{x}, \mathbf{y}) \right] \geq \alpha \right\}, \tag{2}$$

where $L_\mathbf{x}$ is the length of the oracle solution $\mathbf{z}$ for prompt $\mathbf{x}$. Since $i^*(\mathbf{x})$ is defined per prompt, we compute these indices in a pre-processing stage before RL training. Using this, we construct an augmented set of hard prompts:

$$\mathcal{D}_{\text{hard}}^{\text{aug}} := \left\{ \text{concat}(\mathbf{x}, \mathbf{z}^{0:i^*(\mathbf{x})}) \mid \mathbf{x} \in \mathcal{D}_{\text{hard}} \right\}. \tag{3}$$

POPE then trains on a dataset consisting of a 1:1 mixture of hard prompts $\mathcal{D}_{\text{hard}}$ and their augmented versions $\mathcal{D}_{\text{hard}}^{\text{aug}}$. As we show, while unaugmented prompts do not yield reward early in training, the augmented prompts do, and these gains eventually transfer back to the unaugmented prompts, especially when the augmented prompt uses the minimal prefix (Eq. 2). This enables POPE to learn effectively on hard prompts. Finally, we emphasize that POPE operates fully on-policy: although privileged information guides exploration, the exploration itself is carried out by the model through on-policy rollouts. This procedure does not utilize interventions, off-policy corrections, or off-policy completions during RL, keeping the training procedure simple to implement yet effective at learning.

> **Summary: Privileged On-Policy Exploration (POPE)**
>
> - POPE conditions on partial solutions from an oracle as privileged information to guide on-policy rollouts during RL training, instead of directly using oracle data as training targets.
> - POPE utilizes the smallest prefix of the oracle solution that attains reward $\geq \alpha$ for maximal transfer to unaugmented prompts during RL and best performance.

## 6 EXPERIMENTAL EVALUATION

We evaluate the efficacy of POPE in providing training signal on hard problems, and how training on such hard problems translates to test performance. We also wish to understand if POPE alleviates the ray interference issue, and whether the design choices introduced above are crucial.

**POPE implementation.** To instantiate POPE, we begin by constructing a dataset of hard training problems on which the base model fails completely. To do this, we evaluate the base model (e.g., Qwen3-4B) with a rollout budget of $k = 32$ samples per problem. We mark a problem as hard if the model achieves zero success rate, meaning no single rollout produces the correct final answer. For each hard problem, we derive the minimal guidance prefix from an oracle solution that can help the base model obtain non-zero reward. Oracle solutions are obtained either from human-written data or from stronger models such as Gemini 2.5 Pro in our experiments. To identify candidate prefixes, we first chunk each solution at meaningful logical points, for example, individual steps of a derivation, applications of lemmas or theorems, or transitions between reasoning steps. Each chunk boundary provides for a boundary of the potential prefix. We then evaluate the performance of the base model (Qwen3-4B model in our setting) conditioned on these prefixes, checking whether at least one rollout yields the correct answer. We use a token budget of 16K tokens for this evaluation, since this is also the token budget that we run RL training at subsequently. The minimal privileged prefix $i^*(\mathbf{x})$

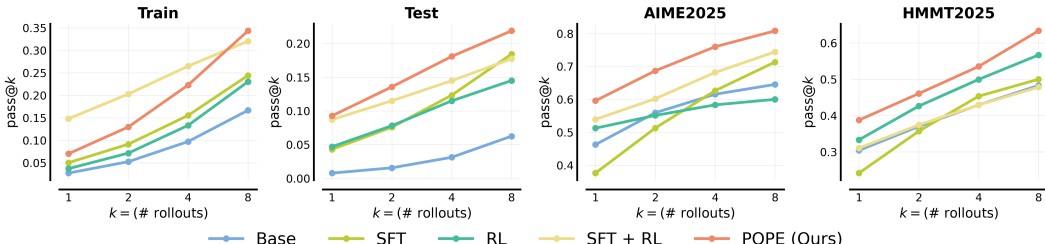

Figure 5: *Pass@k performance of different approaches*, using Gemini 2.5 Pro as the oracle. Importantly, we do not assume access to Gemini's reasoning traces, only a prefix of the final solution as privileged information. POPE achieves the best pass@k performance overall on both the i.i.d. held-out test set of hard problems and the standardized AIME 2025/HMMT 2025 benchmarks. On the training set, POPE also improves pass@8, although SFT+RL (yellow) outperforms it. This is expected since SFT+RL leverages large-scale rejection sampling to obtain correct traces by combining oracle-provided prefixes with base LLM rollouts conditioned on them.

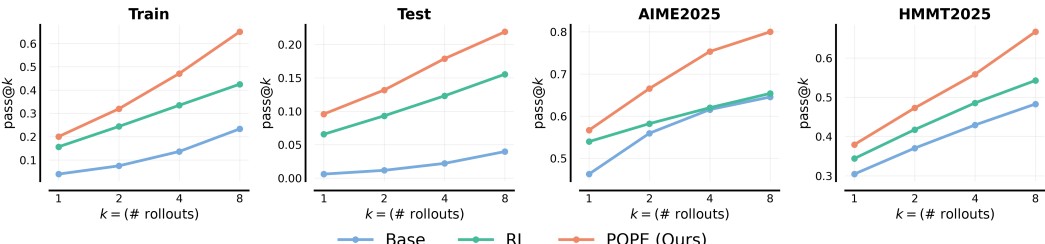

Figure 6: *Pass@k performance of different approaches*, using human solutions as the oracle on Omni-MATH. Observe that POPE outperforms running RL without any form of privileged information, as well the base model. These gains are observed both on the train and test sets, as well as AIME2025 and HMMT2025 benchmarks.

(Equation 2) for a problem $\mathbf{x}$ is defined as the shortest prefix of the oracle solution that leads to some non-zero success on the problem. In our experiments, we choose a threshold success rate of 0.3 over 16 rollouts conditioned on a partial oracle prefix to define this minimal prefix $i^*(\mathbf{x})$. In cases where no prefix leads to success, we still select one prefix at random, ensuring every problem is paired with at some sort of a partial solution from the oracle. While these random prefixes do not immediately induce correct rollouts, they may still facilitate exploration and allow the model to evolve useful behaviors during training. This process yields a dataset of hard problems paired with either minimal or fallback prefixes, providing structured yet lightweight privileged information for exploration. In this work, we train on 495 problems when privileged prefixes are derived from a stronger model, and on 165 problems when they are derived from human-written solutions.

**Evaluation protocol.** We evaluate whether POPE enables models to learn on hard problems during RL. To this end, we measure pass@k performance for $k \in \{1, 2, 4, 8\}$, using 32 evaluation rollouts per prompt to obtain statistically reliable estimates on the training dataset (without conditioning on partial solutions for evaluation). We further assess generalization by evaluating accuracy on a held-out test set drawn from the same distribution as the training set, consisting of problems of comparable difficulty where the base model has a low success rate. Finally, we report performance on standardized competition benchmarks, including AIME2025 and HMMT2025 (Balunović et al., 2025). Together, these metrics allow us to compare approaches on the optimization of training rewards, generalization to problems similar to the training distribution, and also to external benchmarks.

**Comparisons and baselines.** We compare POPE against the following baselines: **(1) Standard RL**, where we run RL on the same prompt set as POPE but without any privileged oracle information for conditioning and thus provides a representative RL training algorithm; **(2) SFT**, where we fine-tune on the complete oracle solution for each hard training problem. This approach corresponds to the default approach for using privileged information as a training target. We also experimented with running RL on top of such SFT checkpoints, but found that these models—particularly when trained on human data—memorized the training data to quite a large extent, making them poor initializations for RL training; and **(3) SFT+RL**, where the base LLM is first supervised fine-tuned on oracle privileged information concatenated with successful reasoning traces generated by the base model when conditioned on that information, and then further trained with RL. This setup resembles a "mid-trained" initialization, similar to rejection fine-tuning (RFT) (Zelikman et al., 2022; Yuan et al., 2023), where privileged information is used to construct RFT data but is not employed during subsequent RL training. Comparing with approach **(1)** enables us to establish the role of oracle

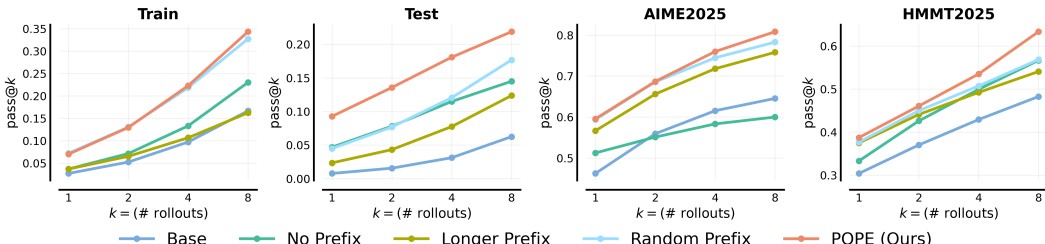

Figure 7: *Ablation on the amount of privileged information used for conditioning.* POPE with minimal prefixes achieves the best performance across train, test, and AIME/HMMT benchmarks. Random prefixes perform comparably on train and AIME 2025 but lag on test and HMMT 2025. Providing longer prefixes is detrimental, as it exacerbates the rich-gets-richer effect and slows progress compared to standard RL.

privileged information for learning in the first place; **(2)** establishes the benefits from conditioning over using privileged information as training targets; and **(3)** allows us to particularly establish the role of on-policy exploration (as opposed to off-policy RFT) for exploration.

## 6.1 MAIN PERFORMANCE RESULTS

**POPE with stonger LLM solutions.** Figure 5 shows results when privileged information for POPE and other methods is provided by Gemini 2.5 Pro. We do not assume access to Gemini's reasoning traces, only a short prefix of its final solution. In this setting, POPE consistently achieves the best pass@$k$ performance on both the held-out test set of hard problems and on standardized math benchmarks such as AIME2025 and HMMT2025. On the training set, POPE also improves pass@8, although SFT+RL can outperform it because it leverages large-scale rejection sampling that combines oracle-provided prefixes with model rollouts. This comparison highlights the importance of on-policy exploration guided by privileged information as opposed to passively distilling it into a model. Overall, we find that prefixes of solutions from a stronger LLM can substantially improve exploration on hard problems. All reported numbers are computed with 32 rollouts.

**POPE with human-written solutions.** Figure 6 shows results using prefixes of human-written solutions from Omni-MATH (Gao et al., 2024) difficulty 5–8 problems as privileged information. Many of these training and test problems are substantially harder than those in the Gemini setting, to the point that even strong proprietary LLMs fail on them. Nevertheless, POPE again outperforms baselines across train, test, and benchmark evaluations. The gains are especially pronounced over standard RL, which is unable to make progress in this regime. These findings demonstrate that POPE can leverage even human solutions to unlock RL training on problems beyond the reach of current LLMs. Taken together with the Gemini setting, these results illustrate the overall efficacy of POPE in improving exploration and performance on harder problems.

## 6.2 ABLATION STUDIES

Next, we present a series of ablation experiments for POPE to better understand the role of different design choices in our approach and in the baselines. We focus on three key questions: **(1)** How does the amount of privileged information influence the efficacy of POPE? **(3)** How do different sources of privileged information, such as a stronger language model (Gemini 2.5 Pro in our case) and human data, compare in their effectiveness? **(3)** To what extent does POPE address the ray interference issue outlined in Figure 2 with standard RL? and **(4)** Does POPE enable transfer of strategies learned under privileged guidance to the original prompt without guidance? We address the first three questions below and provide a qualitative example for answering **(4)**.

**1) Amount of privileged information.** To answer the first question, we ablated POPE by varying the amount of privileged information used for conditioning. Specifically, we compared POPE against: **a)** a *longer prefix*, where the first ¼ of the oracle solution was provided for every problem, and **b)** a *random prefix*, where a randomly sampled prefix of the oracle solution was used as privileged information. As shown in Figure 7, using the minimal prefix prescribed by POPE performs best across the train set, test set, and the AIME/HMMT benchmarks. The random prefix strategy generally performs second best, roughly matching POPE on the train set and AIME 2025, but falling behind on the test set and HMMT 2025. In contrast, providing excessive privileged information through the longer prefix was detrimental: the percentage of unsolved problems decreased no faster than in standard RL without any privileged information. We suspect this is because longer prefixes exacerbate the rich-gets-richer problem: by making some problems substantially easier by revealing a longer prefix than what is needed to attain a particular accuracy under the base model, this strategy can induce a disparate distribution over prompt difficulty in the training set.

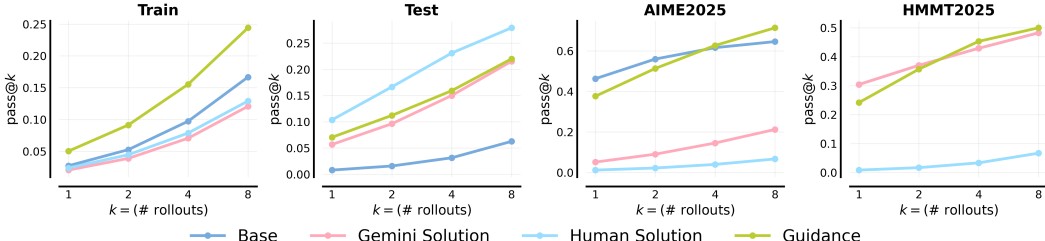

Figure 8: *Comparison of different sources of oracle data as training target.* Cloning reasoning traces generated from the base model itself generated by running rejection sampling on the prompt augmented with privileged prefixes outperforms cloning human or Gemini solutions. However, the base model often performs better than most SFT variants, especially on train, test, and at larger $k$ for AIME and HMMT. This shows that directly using oracle solutions as training targets can degrade pass@$k$ performance.

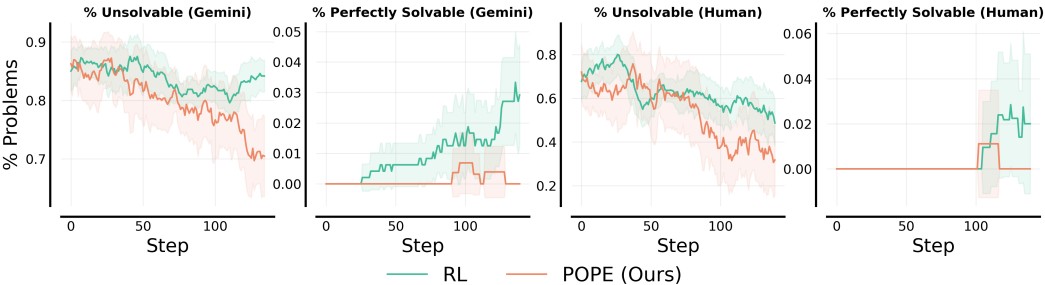

Figure 9: *Assessing ray interference with POPE.* We plot the percentage of unsolved and perfectly solved problems over the course of training, evaluated without privileged information. Naïve RL exhibits the same failure mode as in Figure 3: it initially solves new problems but soon over-sharpens on a subset of prompts, and eventually even unlearns problems solved earlier in training. In contrast, POPE steadily reduces the fraction of unsolved problems and avoids excessive sharpening, thereby mitigating ray interference.

**2) Comparing different sources of oracle data.** We next compare the efficacy of different types of oracle data when used only for supervised fine-tuning (SFT). For this experiment, we fine-tune the base model on three sources: **a)** human-written solutions, **b)** language model generated solutions from Gemini, and **c)** reasoning traces generated by prompting the base model with a prefix of the Gemini solution (the same setup used in our SFT+RL comparison). As shown in Figure 8, SFT with approach **c)**, which clones reasoning traces largely obtained through rejection sampling from the base model, outperforms cloning either human or Gemini solutions. However, we also find that in many cases the base model itself achieves higher performance than any SFT variant, particularly on the train set, the test set, and at larger $k$ values for AIME and HMMT. This result is consistent with prior observations in synthetic data (Setlur et al., 2024) and reinforces our motivation: directly using oracle solutions as training targets is not effective, since it often reduces pass@$k$ performance.

**3) Assessing ray interference with POPE.** Finally, we assess the extent of ray interference under POPE by plotting the percentage of unsolved and perfectly solved problems over the course of training in Figure 9, analogous to Figure 2. Although training uses a mixture of augmented and unaugmented prompts, evaluation is performed without any privileged information. As shown in the figure, naïve RL fine-tuning follows the same failure mode as in Figure 2: the model initially learns to solve new problems, but soon begins to sharpen on a subset of prompts where it consistently produces correct rollouts, while progress on other problems stalls. In fact, as training goes on, the model unlearns certain problems that it could solve earlier in training. In contrast, POPE steadily reduces the fraction of unsolved problems throughout training and avoids excessive sharpening (perfectly solved problems is lower for POPE), thereby addressing the challenge of ray interference.

## 7 DISCUSSION AND PERSPECTIVES ON FUTURE WORK

We introduce POPE: a framework for enabling reinforcement learning on hard reasoning problems by guiding exploration with minimal oracle prefixes. We show that POPE consistently outperforms standard RL and SFT baselines, improves pass@k on both training and held-out test problems, and achieves state-of-the-art results on math benchmarks such as AIME 2025 and HMMT 2025, while mitigating ray interference. Looking forward, we envision extending POPE to multimodal reasoning tasks and adaptive prefix selection strategies that further automate privileged guidance.

## 8 ETHICS STATEMENT

Our training and empirical evaluation centers on mathematical reasoning tasks (e.g., AIME, HMMT, Omni-MATH), which are widely regarded as low-risk, since they rely on publicly available benchmarks without private or sensitive personal data. The risk of producing toxic or harmful outputs in these structured reasoning tasks is minimal relative to open-ended generation. That said, we recognize the broader ethical implications of advancing LLM capabilities. While enhanced reasoning holds significant promise for scientific and educational progress, it also entails dual-use risks, such as potential misuse in generating sophisticated disinformation.

## 9 REPRODUCIBILITY STATEMENT

We have taken several steps to ensure reproducibility of our results. The full description of the `POPE` algorithm is given in Sec. 5, with training details and datasets described in Sec. 6. Training hyperparameters for both supervised fine-tuning and RL are reported in Appendix A (Tables 1 and 2). Qualitative examples of rollouts with and without guidance are provided in Figs. 2, 7. Together, these resources should allow independent researchers to replicate and extend both the empirical and methodological contributions of this work.

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

# Appendices

## A  TRAINING HYPERPARAMETERS

### A.1  HYPERPARAMETERS FOR SFT

For `POPE` and RL runs, we utilize the TRL codebase. The base models are directly loaded from Hugging Face: Qwen3-4B

| Hyperparameter | Values |
|---|---|
| learning_rate | 1.0e-5 |
| num_train_epochs | 3 |
| batch_size | 128 |
| gradient_checkpointing | True |
| max_seq_length | 16384 |
| bf16 | True |
| num_gpus | 8 |
| warmup ratio | 0.1 |

Table 1: Hyperparameters used for `POPE`

### A.2  HYPERPARAMETERS FOR RL

We utilize the verl codebase to run GRPO. We use Qwen3-4B as the base model for training.

| Hyperparameter | Values |
|---|---|
| max_prompt_length | 2048 |
| max_response_length | 16384 |
| clip_ratio_low | 0.2 |
| clip_ratio_high | 0.35 |
| train_batch_size | 32 |
| learning_rate | 1.0e-6 |
| kl_loss_coef | 0.001 |
| temperature | 0.8 |
| critic_warmup | 0 |
| total_training_steps | 300 |
| num_gpus | 16 |

Table 2: Hyperparameters used for `POPE` & RL

## B  USE OF LARGE LANGUAGE MODELS

We used large language models (LLMs) as an assistive tool primarily for rephrasing arguments more crisply and for generating LaTeX templates (*e.g.*, tables, algorithm boxes, or figure formatting). All research ideas, developments, experiments, and empirical results were conceived, executed, and validated by the authors. The LLM did not contribute to the scientific content, claims, or findings of this work.

## C  PROMPTS

We use the following prompt to guide the model in solving a problem with a partial solution.

> **Prompt for Solving with Partial Solution**
>
> ```
> You are given a problem and a partial solution.  Your task is to
>
> carefully study the partial response, identify what reasoning
> or steps are already provided, and then complete the solution
> from where it left off.  Ensure your continuation is logically
> consistent and leads to a complete and correct final answer.
> **Important**:  Show your reasoning step-by-step, and clearly
>
> present the final answer using LaTeX-style \boxed{} notation.
>
> Problem:  <Problem>
> Partial Response:  <Partial Response>
>
> Continue solving the problem, starting from where the partial
> response ends.  Make sure your final answer is written as:
> ```
> $\boxed{< Answer >}$

## D  EXAMPLES

Here we provide an example from the Omni-MATH dataset with a human solution.

> **Question from Omni-MATH**
>
> ```
> Let k ≥ 2 be an integer.  Find the smallest integer n ≥ k + 1 with
> the property that there exists a set of n distinct real numbers
> such that each of its elements can be written as a sum of k other
> distinct elements of the set.
> ```

> **Human Solution**
>
> Let $k \geq 2$ be an integer.  We need to find the smallest integer $n \geq k + 1$ such that there exists a set $S$ of $n$ distinct real numbers, where each element of $S$ can be expressed as a sum of $k$ other distinct elements of $S$.
> To solve this problem, we consider the construction of such a set $S$.
> 1.  **Understanding the Problem:**
> – For each element $s \in S$, we need $k$ distinct elements from $S \setminus \{s\}$ that sum up to $s$.
> 2.  **Minimum Size Construction:**
> – We start by proving that with $n = k + 4$, such a set can indeed be constructed.
> – Consider a construction where:
> – Choose $k + 1$ elements as the base set:  $\{a_1, a_2, \ldots, a_{k+1}\}$.
> – Introduce an additional four elements:  $\{b_1, b_2, b_3, b_4\}$.
> – We construct our set $S$ as:
>
> $$S = \{a_1, a_2, \ldots, a_{k+1}, b_1, b_2, b_3, b_4\}$$
>
> 3.  **Illustrating the Construction:**
> – Arrange the elements such that:
> – Each $a_i$ is expressed as the sum of any $k$ of the other $a_j$'s and some $b$'s if necessary.
> – Each $b_i$ can be expressed using a combination of $a$'s and other $b$'s.
> 4.  **Verification:**

```
   – By choosing specific numbers for each bᵢ, we ensure that each
   number in the constructed set can indeed be expressed as a sum of
   k distinct others.
   – For example, by choosing values and testing that the sum condition
   holds, we verify that each possibility works, fulfilling the
   problem's conditions.
   5.  **Conclusion:**
   – Testing smaller n for valid configurations will fail due to
   insufficient numbers to formulate each possible sum using k
   distinct numbers.
   – Therefore, the smallest n for which such a configuration is
   possible indeed turns out to be n = k + 4.
   Thus, the smallest integer n such that a set S with the given
   conditions can be constructed is:   | k + 4 |
```

Here we provide an example from the DAPO dataset with a solution generated by gemini-2.5-pro.

### Question from DAPO

Find the smallest natural number $n$ such that the expression $n^2 - n + 11$ has exactly four prime factors, which may not be distinct.

### Solution generated by Gemini

Let the given expression be $f(n) = n^2 - n + 11$. We are looking for the smallest natural number $n$ such that $f(n)$ has exactly four prime factors, including multiplicity. This means $\Omega(f(n)) = 4$, where $\Omega(k)$ is the total number of prime factors of $k$.
Let's analyze the expression $f(n) = n(n-1) + 11$. Since $n(n-1)$ is always an even number, $f(n)$ is always odd. So, 2 cannot be a prime factor of $f(n)$. Let's check for other small prime factors:
– Modulo 3: $n(n-1)$ can be $0 \times (-1) \equiv 0$, $1 \times 0 \equiv 0$, or $2 \times 1 \equiv 2$. So $n^2 - n + 11 \equiv n(n-1) + 2 \pmod{3}$ can be $0 + 2 = 2$, $0 + 2 = 2$, or $2 + 2 = 4 \equiv 1$. So $f(n)$ is never divisible by 3. – Modulo 5: $n(n-1) + 11 \equiv n(n-1) + 1 \pmod 5$. $n(n-1)$ can be $0, 2, 6 \equiv 1, 12 \equiv 2, 20 \equiv 0$. So $n(n-1) + 1 \pmod 5$ can be $1, 2, 3$. So $f(n)$ is never divisible by 5. – Modulo 7: $n(n-1) + 11 \equiv n(n-1) + 4 \pmod 7$. $n(n-1)$ can be $0, 2, 6, 12 \equiv 5, 20 \equiv 6, 30 \equiv 2, 42 \equiv 0$. So $n(n-1) + 4 \pmod 7$ can be $4, 6, 10 \equiv 3, 9 \equiv 2$. So $f(n)$ is never divisible by 7. – Modulo 11: $f(n) = n^2 - n + 11 \equiv n(n-1) \pmod{11}$. $f(n)$ is divisible by 11 if and only if $n(n-1) \equiv 0 \pmod{11}$. Since 11 is prime, this occurs when $n \equiv 0 \pmod{11}$ or $n \equiv 1 \pmod{11}$.
So, the smallest possible prime factor of $f(n)$ is 11. We want $f(n)$ to be a product of four primes, $p_1 p_2 p_3 p_4$. To find the smallest $n$, we should look for the smallest possible value of $f(n)$. The function $f(n) = n^2 - n + 11$ is increasing for $n \geq 1$. So smaller values of $n$ give smaller values of $f(n)$.
The smallest number that is a product of four primes, where each prime is at least 11, is $11^4$. Let's see if $f(n)$ can be equal to $11^4$. $11^4 = 14641$. We need to solve $n^2 - n + 11 = 14641$ for a natural number $n$. $n^2 - n - 14630 = 0$. Using the quadratic formula, $n = \frac{-(-1) \pm \sqrt{(-1)^2 - 4(1)(-14630)}}{2(1)} = \frac{1 \pm \sqrt{1 + 58520}}{2} = \frac{1 \pm \sqrt{58521}}{2}$. For $n$ to be an integer, 58521 must be a perfect square. Let's estimate $\sqrt{58521}$. $200^2 = 40000$, $250^2 = 62500$. $240^2 = 57600$. $241^2 = (240 + 1)^2 = 57600 + 480 + 1 = 58081$. $242^2 = (240 + 2)^2 = 57600 + 960 + 4 = 58564$. Since $241^2 < 58521 < 242^2$, 58521 is not a perfect square. Thus, there is no integer $n$ for which $f(n) = 11^4$.
Let's find the next smallest number composed of four prime factors, each at least 11. The candidates for the smallest such numbers are $11^4 = 14641$, $11^3 \times 13 = 1331 \times 13 = 17303$, $11^2 \times 13^2 = 121 \times 169 = 20449$, etc. The

```
smallest is 11⁴, which we've ruled out.  The next smallest is 11³ × 13.
Let's check if f(n) can be equal to 11³ × 13 = 17303.  We need to solve
n² − n + 11 = 17303.  n² − n − 17292 = 0.  Using the quadratic formula,
n = (1±√(1−4(1)(−17292)))/2 = (1±√(1+69168))/2 = (1±√69169)/2.  For n to be an integer,
69169 must be a perfect square.  Let's estimate √69169.  200² = 40000,
300² = 90000.  260² = 67600.  The last digit is 9, so the root could end
in 3 or 7.
Let's try 263² = (260 + 3)² = 67600 + 2 × 3 × 260 + 9 = 67600 + 1560 +
9 = 69169.  So, 69169 is a perfect square, and its square root is 263.
Then n = (1±263)/2.  This gives two possible integer values for n:  n₁ =
(1+263)/2 = 264/2 = 132.  n₂ = (1−263)/2 = −262/2 = −131.  Since we are looking for a
natural number n, we have n = 132.  For n = 132, f(132) = 132² − 132 + 11 =
17424 − 132 + 11 = 17292 + 11 = 17303.  And 17303 = 11³ × 13.  The prime factors
are 11, 11, 11, 13.  There are exactly four prime factors.  So, n =
132 is a candidate.
To be sure that n  =  132 is the smallest such natural number, we
need to confirm that no smaller n results in a four-prime-factor
number.  The function f(n) is increasing for n  ≥  1.  So for any
n  <  132, f(n)  <  f(132) = 17303.  If there were a smaller n₀  <  132
such that f(n₀) has four prime factors, then f(n₀) would have to be
a number smaller than 17303 with four prime factors, each being at
least 11.  The smallest such number is 11⁴  =  14641.  We have already
shown that f(n) cannot be 14641.  The next smallest such number is
11³ × 13 = 17303, but this value is not smaller than 17303.  Any other
combination of four primes (e.g., 11² × 13²  =  20449) will be larger.
Therefore, there is no value of f(n) with four prime factors smaller
than 17303.  This means that no n  <  132 can produce a value of f(n)
with four prime factors.  Thus, the smallest natural number n is
132.
Final check:  For n  =  132, f(132)  =  132² − 132 + 11  =  17303.  Prime
factorization of 17303:  17303 = 11 × 1573  1573 = 11 × 143  143 = 11 × 13  So
f(132) = 11 × 11 × 11 × 13 = 11³ × 13.  This number has 3+1 = 4 prime factors.
```

Here we provide example rollouts from a model fine-tuned with POPE.

**Figure 10:** *Example rollouts from a model fine-tuned with POPE,* shown with and without conditioning on a partial oracle solution. In both cases the model adopts a similar strategy: when guidance is provided it follows the partial solution, and when guidance is absent it independently proposes the same strategy. This illustrates at least some form of knowledge sharing across augmented and unaugmented versions of the prompts.

