# OpenReview forum: "Learning to Reason on Hard Problems with Privileged On-Policy Exploration"
_ICLR.cc/2026/Conference — Submitted to ICLR 2026_

### Official Review · Reviewer_b7vm · 2025-10-28

**Soundness:** 2
**Presentation:** 3
**Contribution:** 2
**Rating:** 4
**Confidence:** 4

**Summary:**

This paper addresses the exploration limitation of reinforcement learning (RL) for reasoning in large language models (LLMs),  to be more specific, the inability of on-policy methods to learn from hard problems that yield zero reward signals. Standard RL fine-tuning (e.g., GRPO) often overfits to “easy” problems, sharpening its policy distribution and reducing exploration diversity, leading to a phenomenon the authors term ray interference: gradients from easy problems suppress progress on unsolved ones.

To overcome this, the authors propose POPE (Privileged On-Policy Exploration), a method that leverages privileged information (prefixes of correct solutions from human or model “oracles”)  as guidance for on-policy rollouts. Instead of using such traces for distillation or supervised fine-tuning, POPE conditions on minimal solution prefixes that allow a base model to achieve non-zero reward on otherwise unsolvable problems. RL training is then performed on a 1:1 mixture of original and augmented prompts, enabling learning signals to transfer back to unaugmented (hard) problems.

**Strengths:**

- well-motivated research question
- empirical experiments show its strength and improved exploration

**Weaknesses:**

- limited theoretical formalization. While ray interference is discussed qualitatively, the paper lacks a theoretical model explaining why POPE’s conditioning mitigates it
- dependence on external oracles. POPE requires human-written or high-quality LLM solutions for prefix extraction.

**Questions:**

First of all, I would like to thank the authors for their work. I agree that reinforcement learning (RL) post-training for large language models (LLMs) tends to sharpen output distributions, and I appreciate that the paper tries to address this important research question.

Here are some concerns that I have.

- lack of theoretical analysis of why POPE’s conditioning mitigates the exploration bottleneck.

- Do the observed benefits primarily stem from directly providing partially correct answers? Alternatively, what would happen if the model were given partially incorrect answers instead? Would its performance then be limited to the correctness of the provided inputs?

- Would enhancing the exploration capability of the RL method similarly improve exploration behavior during RL post-training, as demonstrated by POPE?  Furthermore, could the observed sharpening of the output distribution, where the model converges to simpler, intermediate tasks, be interpreted as a form of reward hacking within the RL process? If so, could reward reshaping be used to mitigate this effect?

---

### Official Review · Reviewer_9n4z · 2025-10-28

**Soundness:** 2
**Presentation:** 3
**Contribution:** 2
**Rating:** 4
**Confidence:** 3

**Summary:**

This paper targets an important RL issue ehich is it cannot use all training problems in a training dataset. Too hard or too easy samples all cause some issues. This paper introduces a framework that leverages already available solutions to build a minimal solution prefix as guidance to obtain a learning signal on hard problems.

**Strengths:**

1. The paper focuses on an interesting and important question
2. The motivation and intuition behind the method are clear.
3. The method shows clear improvement.

**Weaknesses:**

1. My main concern lies in the novelty of the paper. From my understanding, both the motivation and the proposed approach are closely related to prior works such as [1,2]. In particular, [2] also employs a minimal-length prefix to obtain positive rewards. Therefore, the contribution of this work appears incremental relative to these recent studies.
2. The current experiments are limited to a single base model. Evaluating the proposed method across multiple base models, varying in both size and architecture family, would provide stronger evidence of its robustness and general applicability.
3. The prefix length is computed in a pre-processing stage before RL training. But the minimal prefix that allows on-policy rollouts to obtain some non-zero reward will change during the training. With more training steps, the model gets better and better, it may need a shorter prefix.
4. The model is trained on a dataset consisting of a 1:1 mixture of hard prompts and their augmented versions. Does it have to be 1:1? Is there any exploration of changing the propotion?
5. Minor: There might be a type, in section 6.2, it list three key questions (1) (3) (3) (4).
[1] Amani, Mohammad Hossein, et al. "RL for Reasoning by Adaptively Revealing Rationales." arXiv preprint arXiv:2506.18110 (2025).

[2] Zhang, Xuechen, et al. "BREAD: Branched Rollouts from Expert Anchors Bridge SFT & RL for Reasoning." arXiv preprint arXiv:2506.17211 (2025).

**Questions:**

See weakness

---

### Official Review · Reviewer_Jq3v · 2025-10-29

**Soundness:** 4
**Presentation:** 4
**Contribution:** 3
**Rating:** 6
**Confidence:** 2

**Summary:**

This work IS motivated by the observation that when applying RL to LLMs, the model will get stuck on the hard problems since no successful trajectories can be found. To address this challenge, they introduce a framework Privileged On-Policy Exploration (POPE) that utilizes partial oracle solutions to increase the initial success rate such that RL converges faster.

**Strengths:**

1. The paper is well-written and well-organized.
2. Experimental evaluations cover multiple aspects of the proposed method.

**Weaknesses:**

1. Literature review is not sufficient. There are several recent works that attempt to improve RL training for LLMs using (a) hints generated by oracle models, (b) partial SFT trajectories, (3) etc. However, this paper neither discusses these related work nor includes experimental comparisons with them.
2. Calculating $i*$ in Eq (2) is costly, since it requires generating many rollouts for each single $i$ and $x$ to get the average reward.

**Questions:**

1. In Eq (2), for a fixed $x$, is $z$ unique and the goal is to find the minimal $i$ of this fixed $z$, or can $z$ be randomly generated each time?
2. How does $i*$ change across different problems?
3. How does such method affect the performance on the easier problems where oracle hints might be unnecessary?

---

### Official Review · Reviewer_22pR · 2025-10-30

**Soundness:** 2
**Presentation:** 2
**Contribution:** 2
**Rating:** 2
**Confidence:** 4

**Summary:**

This paper proposes Privileged On-Policy Exploration (POPE), a method to use partial solution as hint to help RL training on hard reasoning problems.

**Strengths:**

1. Paper writing is clear and fluent.
2. The findings are practical and useful.
3. Experiment shows the effectiveness of POPE on mathematical tasks.

**Weaknesses:**

1. Novelty concern: it seems that this method is too similar to "BREAD: Branched Rollouts from Expert Anchors Bridge SFT & RL for Reasoning", which has already been published.
2. The model size, model family, and reasoning task diversity in the experiment section are limited.

**Questions:**

1. Could you please state more contributions on the method/algorithm side?
2. Could you please include more model families (Deepseek, Llama, etc.), more sizes of models, and some other reasoning tasks (coding, commonsense reasoning, etc.) in the experiment section?
3. Could you please add some analysis about the training stability and method generalization ability, such as out-of-domain performance?

---

### Meta-Review · Area_Chair_Jjm1 · 2025-12-26

**Summary:**

This paper proposes Privileged On-Policy Exploration (POPE), a reinforcement learning framework for improving LLMs’ reasoning on hard problems. POPE leverages partial solution prefixes provided by human or model oracles to obtain a learning signal on hard problems by using them as "privileged" information that guides exploration. Experiments show that POPE expands the set of solvable problems and improves performance on challenging reasoning benchmarks.

Reviewers think the paper focuses on an interesting and important problem. The motivation of the paper is clear, the writing is generally well-organized, and the empirical results demonstrate consistent performance improvements on mathematical reasoning benchmarks. Besides, the findings of the paper are practical and useful.

The main concerns raised by reviewers focus on limited novelty. Various reviewers have mentioned that the motivation and proposed methods of this paper are closely related to existing work, especially BREAD. Furthermore, the related work in this paper is not comprehensive enough. Many works have already attempted to use hints generated by oracle models or partial SFT trajectories, but this paper does not discuss these related works or provide experimental comparisons. In addition, reviewers pointed out that the experimental aspects of this paper are insufficient; for example, experiments were only conducted on a single base model. Furthermore, the dependence on external oracles introduces additional constraints.

During the rebuttal process, the authors did not respond to the reviewers' comments; therefore, it cannot be considered that the reviewers' concerns have been addressed. Considering the reviewers' comments, the contribution does not meet the bar for acceptance. I recommend rejection.

**Reviewer Concerns:**

The authors did not respond to the reviewers' comments during the rebuttal. I think some key concerns have not been resolved, including 1) limited novelty, 2) Insufficient literature review, 3) Insufficient experiments (single base model, comparison with related works), and 4) dependence on external oracles.

**Reviewer Scores:**

Four reviewers submitted their reviews: one recommended reject (score 2), two recommended weak reject (score 4), and one recommended weak accept (score 6). The authors did not respond to the reviewers' comments during the rebuttal.

---

### Decision · Program_Chairs · 2026-01-26

Reject